# Significant Advancements and Evolutions in Chimeric Antigen Receptor Design

**DOI:** 10.3390/ijms252212201

**Published:** 2024-11-13

**Authors:** Anna Gaimari, Anna De Lucia, Fabio Nicolini, Lucia Mazzotti, Roberta Maltoni, Giovanna Rughi, Matteo Zurlo, Matteo Marchesini, Manel Juan, Daniel Parras, Claudio Cerchione, Giovanni Martinelli, Sara Bravaccini, Sarah Tettamanti, Anna Pasetto, Luigi Pasini, Chiara Magnoni, Luca Gazzola, Patricia Borges de Souza, Massimiliano Mazza

**Affiliations:** 1Scientific Institute for Research, Hospitalization and Healthcare, IRCCS Istituto Romagnolo per lo Studio dei Tumori (IRST) “Dino Amadori”, 40121 Meldola, Italy; anna.gaimari@irst.emr.it (A.G.); anna.delucia@irst.emr.it (A.D.L.); fabio.nicolini@irst.emr.it (F.N.); lucia.mazzotti@irst.emr.it (L.M.); roberta.maltoni@irst.emr.it (R.M.); matteo.zurlo@irst.emr.it (M.Z.); matteo.marchesini@irst.emr.it (M.M.); claudio.cerchione@irst.emr.it (C.C.); giovanni.martinelli@irst.emr.it (G.M.); luigi.pasini@irst.emr.it (L.P.); chiara.magnoni@irst.emr.it (C.M.); luca.gazzola@irst.emr.it (L.G.); massimiliano.mazza@irst.emr.it (M.M.); 2Centro Trial Oncoematologico, Department of “Onco-Ematologia e Terapia Cellulare e Genica Bambino” Gesù Children’s Hospital, IRCCS, 00165 Rome, Italy; giovanna.rughi@gmail.com; 3Department of Immunology, Centre de Diagnòstic Biomèdic, Hospital Clínic of Barcelona, 08036 Barcelona, Spain; mjuan@clinic.cat; 4Institut D’Investigacions Biomèdiques August Pi i Sunyer (IDIBAPS), 08036 Barcelona, Spain; dparras@recerca.clinic.cat; 5Faculty of Medicine and Surgery, “Kore” University of Enna, 94100 Enna, Italy; sara.bravaccini@unikore.it; 6Centro Ricerca Tettamanti, Clinica Pediatrica, Università Milano Bicocca, Osp. San Gerardo/Fondazione MBBM, 20900 Monza, Italy; sarahtettamanti5@gmail.com; 7Oslo University Hospital, 0424 Oslo, Norway; anna.pasetto@ki.se; 8Department of Biomedical and Neuromotor Sciences, University of Bologna, 40127 Bologna, Italy

**Keywords:** CAR-T cells, CAR-T therapy, adoptive cell therapy, immunotherapy, ATMP

## Abstract

Recent times have witnessed remarkable progress in cancer immunotherapy, drastically changing the cancer treatment landscape. Among the various immunotherapeutic approaches, adoptive cell therapy (ACT), particularly chimeric antigen receptor (CAR) T cell therapy, has emerged as a promising strategy to tackle cancer. CAR-T cells are genetically engineered T cells with synthetic receptors capable of recognising and targeting tumour-specific or tumour-associated antigens. By leveraging the intrinsic cytotoxicity of T cells and enhancing their tumour-targeting specificity, CAR-T cell therapy holds immense potential in achieving long-term remission for cancer patients. However, challenges such as antigen escape and cytokine release syndrome underscore the need for the continued optimisation and refinement of CAR-T cell therapy. Here, we report on the challenges of CAR-T cell therapies and on the efforts focused on innovative CAR design, on diverse therapeutic strategies, and on future directions for this emerging and fast-growing field. The review highlights the significant advances and changes in CAR-T cell therapy, focusing on the design and function of CAR constructs, systematically categorising the different CARs based on their structures and concepts to guide researchers interested in ACT through an ever-changing and complex scenario. UNIVERSAL CARs, engineered to recognise multiple tumour antigens simultaneously, DUAL CARs, and SUPRA CARs are some of the most advanced instances. Non-molecular variant categories including CARs capable of secreting enzymes, such as catalase to reduce oxidative stress in situ, and heparanase to promote infiltration by degrading the extracellular matrix, are also explained. Additionally, we report on CARs influenced or activated by external stimuli like light, heat, oxygen, or nanomaterials. Those strategies and improved CAR constructs in combination with further genetic engineering through CRISPR/Cas9- and TALEN-based approaches for genome editing will pave the way for successful clinical applications that today are just starting to scratch the surface. The frontier lies in bringing those approaches into clinical assessment, aiming for more regulated, safer, and effective CAR-T therapies for cancer patients.

## 1. Introduction

Recent advances in immunotherapy have supplied oncologists with an unprecedented array of solutions to boost the host immune system against cancer cells, leading to a substantial improvement in disease-related survival, particularly in haematological malignancies. In many cases, the benefits of immunotherapy also include a marked reduction in side effects compared to conventional radiotherapy or chemotherapy [1,2,3].

Standard immunotherapy approaches largely consisted of monoclonal antibodies (mAbs) in immune checkpoint inhibitors (ICIs). Only recently, mRNA vaccines, oncolytic viruses, adoptive cell therapy (ACT), which includes tumour-infiltrating lymphocyte (TIL) and chimeric antigen receptors (CARs) with engineered T cells, have entered the mainstream of cancer therapeutic options [4,5,6]. These infiltrate tumours to achieve long-term remission by uplifting the patient’s immune system to specifically recognise and destroy cancer cells [7]. In particular, ACT harnesses patient-derived immune cells, which are genetically modified to express a synthetic receptor, such as a CAR or a TCR, which targets specific antigens on the surface of cancer cells. The goal is to trigger a cytotoxic response against the tumour [8].

The CAR is a recombinant receptor composed of the antigen-binding domain, typically a single-chain variable fragment (scFv) derived from a monoclonal antibody (mAb), which is molecularly fused to intracellular signalling domains that activate immune cells. The antigens recognised by the CAR are either tumour-specific or tumour-associated antigens. The spacer or hinge is the flexible portion of the CAR that links the extracellular domain to the intracellular domain and is responsible for receptor flexibility in antigen recognition and, to some extent, for the modulation of downstream signalling [9]. The intracellular domain is the functional portion of the receptor, consisting of both signalling and co-stimulatory domains. CD3 is the most commonly used signalling domain that, upon antigen engagement by the CAR, triggers a nuclear factor of the activated T cell (NFAT)-dependent transcriptional pathway. This activation is essential to elicit T cell cytotoxicity. Signalling domains can be combined with costimulatory molecules, such as CD28, 4-1BB, CD27, and OX40, among others. These costimulatory molecules determine efficient receptors’ activation, influencing the cytotoxicity rate and the development of an immunological memory phenotype [9,10]. Leukapheresis of the patient is often the first step in producing autologous CAR-T cells, while some researchers have explored the use of non-separated whole peripheral blood (Figure 1). The T cells are then stimulated and co-stimulated ex vivo in the presence of a cytokine cocktail. In addition, stimulating antibodies can be expressed on artificial antigen-presenting cells, chemically attached to beads, or supplied in soluble form [11]. Before apheresis, patients must abstain from all medications for two weeks to guarantee enough cell counts and quality viability for the production process, which lasts on average between two and four weeks [12]. CAR-T cells are genetically modified and typically generated by introducing a CAR gene into autologous T cells using a retro/lentivirus or transposable elements [13]. Several clinical trials have shown substantial efficacy with those approaches [14]. However, safety concerns have recently been raised on the use of transposon systems, such as PB transposons, which have been associated with the development of CAR-T cell lymphomas in treated patients [15].

To date, six CAR-T cell therapies have been approved by the Food and Drug Administration (FDA), with the first approved back in 2017 [16]. All have received approval for treating refractory and relapsed (r/r) haematological malignancies, namely acute B lymphoblastic leukaemia (B-ALL), B-cell lymphoma (DLBCL, PMBCL), follicular lymphoma (FL), and mantle cell lymphoma (MCL)hematologicalmyeloma (MM). These include (i) tisagenlecleucel (Kymriah™, tisa-cel, Novartis, Basel, Switzerland), an anti-CD19 CAR-T cell therapy with 4-1BB-CD3 domains, manufactured from cryopreserved peripheral blood mononuclear cells (PBMCs); (ii) axicabtagene ciloleucel (Yescarta™, axi-cel, Kite Pharma, Hoofddorp, Netherlands), an anti-CD19 CAR-T construct having CD28-CD3 domains, produced from fresh PBMCs; (iii) brexucabtagene autoleucel (Tecartus™, brex-cel, Kite Pharma), an anti-CD19 CAR-T with CD28-CD3 domains obtained from enriched T cells; (iv) lisocabtagene maraleucel (Breyanzi™, liso-cel, Juno Therapeutics & Bristol Myers Squibb, Seattle, US), an anti-CD19 CAR-T with 4-1BB-CD3 domains, produced as a 50% CD4:CD8 T cell mixture; (v) idecabtagene vicleucel (Abecma^®^, ide-cel, Bristol Myers Squibb), an anti-BCMA with 41BB-CD3 domains produced as a 50% CD4/CD8 CAR-T ratio [16]; and, finally, (vi) Ciltacabtagene Autoleucel (Carvykti^®^), developed to treat relapsed or refractory multiple myeloma (r/r MM) [17,18], a CAR T cell therapy that can bind to two unique BCMA epitopes. This is achieved through a combination of two BCMA-targeting nanobody heavy chains (VHH), a 4-1BB costimulatory domain, and a CD3ζ signalling domain [19].

ACT has proven quite effective for patients affected by haematological tumours like acute lymphoid leukaemia and multiple myeloma [20]. However, in most clinical cases, its efficacy on solid tumours has been limited, since treating solid tumours with ACT poses several challenges. First, solid tumours present an enormous heterogeneity of antigens and antigen expression levels within a single tumour mass [21]. Second, clinicians very often encounter the risk of tumour relapse after targeted therapies, despite initial success in treatment response [20]. Typically, tumour relapse is accompanied by a loss of surface antigens as the malignancy progresses, making antigen-targeting therapies particularly precarious. This increases the risk of off-tumour toxicity to healthy tissues expressing the same antigens, albeit at lower levels [22].

Efforts to overcome this issue have considered selecting more than one CAR target simultaneously. Still, a broader T cell specificity toward multiple targets may also increase the risk of off-tumour toxicity. In addition, the presence of collagen stroma, which typically encloses a solid tumour, significantly hampers T cells’ infiltration, and prevents the success of most CAR-T therapy attempts [23,24]. Moreover, the solid tumours’ immunosuppressive environment further decreases the efficacy of CAR-T cells by inducing T cell functional exhaustion, primarily via increased levels of inhibitory cytokines and the downregulation of interleukin-2 [25,26]. An unfriendly tumour environment also shows increased levels of inhibitory cytokines [27] and the presence of cancer-associated fibroblasts, which promote a dense fibrotic microenvironment that limits the transit of T cells [28]. Additionally, aberrant expression of cell adhesion molecules contributes to restricting the extravasation of T cells from the blood vessels into the tumour mass by creating an unconventional capillarisation [29,30].

Failure of CAR-T therapy against solid malignancies may be attributed to these numerous challenges posed by the molecular and cellular microenvironment. Moreover, both experimental and clinical factors, including disease progression, inadequate T cell harvest, and delays in CAR-T production, may further contribute to the failure of CAR-T therapy [31].

These insights have led to the consensus that treating non-haematological cancers may require more practical and smarter strategies, such as combining CAR-T cell therapy with immune checkpoint inhibitors, chemotherapy, or radiotherapy. This approach aims to create a breeding ground for CAR-T cells by enhancing the host’s adaptive immune response [32]. To effectively target various tumour antigens, regulate CAR-T cell function, and enhance safety and efficacy, CAR-T cells with distinct characteristics and significant potential have been developed [33]. These advanced multi-targeted programmable CARs include split CARs, TRUCKs, universal CARs (UCARs), switchable and programmable universal (SUPRA) CARs, Tandem CARs, and CARs with SynNotch technology. The advantages and construction of these innovative smart CARs are thoroughly detailed in the following sections [34]. Various genome editing approaches, such as CRISPR-Cas9 nucleases, homing endonucleases, zinc finger nucleases (ZFN), meganucleases, and transcription-activator-like effector nucleases (TALENs), have been employed to create transgenic CAR-T cells. These techniques help to boost CAR-T cell specificity, reduce toxicities, and tackle the suppressive effects of the tumour microenvironment (TME) [35]. This review aims to provide a comprehensive overview of research and development in the CAR field, with a specific angle on the molecular makeup/composition of the receptors under investigation.

## 2. A Multitude of CAR Molecules

In recent years, several CAR-T cell engineering strategies have been developed. Figure 2 serves as a visual guide to help the reader navigate the upcoming text, summarising key aspects of CAR-T cell therapy optimisation, including CAR molecule modifications, non-CAR molecule variations, external-stimuli-induced approaches, and the latest in vivo advancements, all of which will be further detailed in the following sections.

### 2.1. CAR-T Generations

Conventionally, five CAR vector generations are recognised in the field of immunotherapy, each representing a distinct evolution of CAR design and engineering. These generations encompass improvements in terms of CAR structure, signalling domains, and targeting capabilities, aiming to enhance the specificity, efficacy, and safety of CAR T cell therapies according to the number and type of co-stimulatory modules and the possibility of expressing specific cytokines [36].

The first generation of CAR-T consists of an extracellular scFv domain connected to an intracellular CD3 signalling domain. The first clinical applications showed a low survival rate and an inability to produce lasting immunological responses. Therefore, second-generation CAR molecules have been developed by adding a costimulatory domain (i.e., CD28 or 4-1BB) to the CD3 signalling domain. The third-generation CAR design encompassed the use of two co-stimulatory domains. Compared to the first and second generations, this type of CAR-T cell exhibited superior efficacy, significant proliferation, and better durability when targeting tumour cells [34,36].

However, third-generation CAR-T therapies, while more advanced than their predecessors, still face significant obstacles. Manufacturing these cells remains a complex, resource-intensive process, which limits scalability and accessibility. Moreover, their clinical effectiveness has not consistently surpassed second-generation CAR-T cells, especially in solid tumours, where the tumour microenvironment further hampers their performance. These limitations underscore the ongoing need for innovation to enhance both their efficiency and therapeutic impact [37].

The fourth generation, also known as TRUCK cells (T Redirected Universal Cytokine Killer cells), is engineered to produce cytokines, enzymes, and other biochemical substances [38]. Common cytokines used in TRUCKs include interleukin (IL)-2, IL-15, and IL-18, which enhance the immune response against solid tumours and extend the persistence of CAR-T cells within the tumour microenvironment [39,40]. Finally, fifth-generation CARs are designed with additional elements into the standard CAR design, enabling them to detect multiple antigens or targets, even in cases of low antigen density, further improving their therapeutic potential [41].

Specific CAR designs and strategies have also emerged to boost specificity, efficacy, and to lower the escape of tumour cells. Some of those strategies comprise logic-gated constructs [42]. Examples of this kind are as follows:“OR logic”: Modular T cells can eliminate different malignant cells with various switching modules that target different antigens on cancer cells. Tandem CAR-T cells, for example, contain two bispecific ligand-binding domains. T cells are activated when one or the other antigen-binding domain binds their antigen on the target cell’s surface.“AND logic”: T cells express two different CARs, activating the immune response only when both antigens are expressed on the target cell. Dual CARs are a practical example of this logic.“NOT logic”: This is used when a specific antigen is expressed both by normal and cancerous cells, and a certain one is expressed only by normal cell/healthy tissue. Two complementary modules, targeting the antigen present only on normal cells, are used so they can inactivate each other. The following chapters summarise the features of different CAR types available and under development, comprising pros and cons for each class, as summarised in Table 1.

### 2.2. Ground-Breaking CAR-T: Universal CAR

Universal CARs (UCARs) possess a split structure where a distinct extracellular domain is coupled with a targeting molecule. In UCARs the extracellular adapter connects a soluble tumour antigen-targeting ligand like, for example, a specific antibody (Figure 2) to the transmembrane receptor [57,73]. The distinctive structure of a UCAR, in contrast to conventional CARs, facilitates efficient and selective targeting of several tumour antigens at once. The extracellular domain of all UCARs binds to an adapter that, carrying a specific antigen-binding domain, serves as a connector with tumour cells. This engagement generates an immunological synapse that results in CAR-T cell activation [58,74]. These new UCARs are being developed to boost adaptability and broaden antigen recognition by dividing the T cell signalling unit and antigen-targeting domain using a “third-party” intermediary system [74,75,76]. With the help of this “third-party” “lock-key” split CAR system, CAR-T cells have the potential to be almost entirely antigen-specific [58].

All currently available CAR-T cell products, on the market or in clinical trials, are autologous (derived from the patient’s own lymphocytes). This approach prevents the risk of severe alloimmune rejection and graft versus host disease (GVHD) caused by major histocompatibility complex (MHC) mismatch between donor and recipient. In contrast, UCAR-T cells are allogeneic, obtained by modifying T cells from healthy donors. While UCAR-T cells share the same killing mechanism as traditional CARs, they differ significantly in their manufacturing process, applications, and costs [77]. The autologous T cell manufacturing platform necessitates a highly qualified and committed facility in addition to a sizable infrastructure investment. Furthermore, the manufacturing process is personalised and time-consuming, delaying the treatment of patients who need urgent therapy. In addition, the risk of production failure is concerning, as it may not be possible to generate a sufficient number of personalised autologous T cells from severely ill patients with significant lymphopenia due to previous chemotherapy, radiation therapy, or stem cell transplantation. This costly and time-consuming approach limits its broad application across different tumour types. On the contrary, UCAR-T cells are generated from healthy allogeneic donors. The key concept behind this approach is to genetically modify the donor T cells by altering their TCR genes and/or HLA class I locus to eliminate the risk of GVHD. Targeting genomic sequences within the constant regions or subunits of the endogenous TCR, or disrupting the HLA-A locus of the MHC gene complex, suppresses the expression of TCR or HLA class I molecules. As a result, T cells become incapable of recognising allogeneic antigens, contributing to GVHD prevention. As detailed in the “genome editing” section below, the clinical exploitation of UCAR-T is indeed tightly linked to the progress of genome editing technology. A variety of positional genome editing strategies and enzymes like ZFNs [78], TALENs [79], and CRISPR/Cas9 [50,80] found an application in the UCAR-T space to prevent unwanted toxicities, graft rejection, GVHD, and enhance the persistence of CAR cells. Those advances generate an off-the-shelf living drug.

The split structure of the UCAR is designed to support and exploit the multiple targeting of antigens from a single CAR, which further increases tumour recognition and specificity and reduces the chance of resistance through tumour antigen escape. Thus, tumour antigen heterogeneity can be overcome by the administration of distinct soluble adaptors of a UCAR [73,75]. The UCAR also has the ability to target new antigens via the soluble adaptor without needing to re-engineer a new specific CAR [48,57]. As previously mentioned, the unrestrained activity of conventional CARs, which continues even after tumour eradication, leads to adverse effects and damage to healthy tissues [49,52]. To bypass this issue, the flexible and manageable structure of UCAR contributes to controlling the CAR-T cell performance by the “on switch” mechanism. Based on this, the CAR-T cells can be activated by the administration of the soluble adaptor and can be inactivated by the withdrawal of the soluble adaptor or administration of the non-specific adaptor. Consequently, UCARs with controllable activity can prevent “on-target/off-tumour” toxicity in healthy tissues [57,74].

Additionally, this flexible CAR structure changes the original rigid structure of the CAR to improve security and feasibility. As a bridge between CAR-T cells and tumour cells, the dosage of the switching module can be titrated because it conforms to general pharmacokinetics, and its affinity to target antigens can be regulated by fine-tuning the structure to take control of CAR-T cell activation. Moreover, CAR-T cells are held back by blocking agents, which competitively inhibit switching modules when necessary [53].

To optimise the UCAR function, tag molecules can be applied to soluble targeting receptors. Considerably, the proximal or distal location of the tag has a crucial role in the recognition and binding of the tumour antigen [48,74]. Moreover, the spacer’s length and flexibility affect the CAR-T cell’s efficacy [54]. Regulating the infusion schedule and dosage are other manageable mechanisms to modulate UCAR-T-based treatments. In this way, the infusion time and dosage of UCAR, with or without the soluble targeting ligand, determine the activation or inactivation of CAR-T cells at the tumour site. Accordingly, CRS and CAR-T cell anti-tumour activity in the tumour site could be better controlled through the appropriate administration of a UCAR and the amount of soluble targeting ligand [54,57]. This controllable regimen does not exist in conventional CAR-T cells, leading to the risk of CRS by overproduction of IL-1β, IL-6, and tumour necrosis factor (TNF)-α cytokines [81,82]. While UCAR-T cells offer several advantages, they also come with certain associated side effects. The administration of the UCAR composed of murine scFv may induce immunogenicity, eliciting a human anti-mouse IgG antibody (HAMA) response against the CAR construct. Clinically, patients who have received CAR-T cell therapy may experience cardiac arrest and anaphylaxis shock due to the presence of HAMA. Thus, to tackle this issue, employing humanised scFvs, instead of murine scFv, should be devised to de-risk HAMA-related toxicities from UCARs. Moreover, the long-term persistence of the UCAR-T cells in the tumour site may lead to “on-target/off-tumour” toxicity, CRS, and neurotoxicity. This hurdle can be overcome with the withdrawal of the soluble ligands [83,84]. Thereby, the dose modification and duration of the soluble ligand of the CAR should be optimised to normalise the duration of UCAR activity in the tumour site, minimising injury to the normal cells [48].

Four UCAR types have been described so far: anti-tag CARs, immunological receptors activated by bispecific proteins, immune receptors mediated by antigen-dependent cell cytotoxicity (ADCC), and immune receptors specific to tags [58].

### 2.3. Anti-Tag-Specific Universal Receptor

The scFv receptor of the UCAR with anti-tag specificity is designed to detect the moieties coupled to the tumour antigen-binding domains. When the CAR-T cell’s anti-tag scFv receptor is engaged to the tumour antigen by ligands or antibodies tagged with the corresponding moiety, the CAR-T cell can trigger tumour cytotoxicity. Currently, anti-tag CARs have been developed against biotin [75], fragment crystallisable gamma (Fcg) [85], nuclear antigen-La-SS-B (E5B9) [86], and fluorescein isothiocyanate (FITC) [59,87,88]. The UCAR’s soluble antigen-binding domain is composed of various molecules, including scFv, nano-bodies, Fabs components, and other small molecules. The versatility of the UCAR system in recognising diverse soluble antigen-binding domains broadens UCAR’s applicability and performance [86,88].

### 2.4. The Tag-Specific Immune Receptor

The UCAR’s tag-specific immune receptor construction is based on ligand–ligand interaction. To create the first universal and modular CAR, which Urbanska et al. proposed in 2012, the natural interaction between biotin and avidin was exploited. In this construct, avidin acts as an extracellular domain bound to T cells’ intracellular and transmembrane domains in so-called biotin-binding immune receptor (BBIR) CAR. Full-length antibodies, scFvs, or other tumour-specific ligands are examples of biotinylated antigen-specific molecules associated with the biotin that can trigger T cell activation via avidin–biotin-specific interaction [75]. Overall, this approach aims to broaden the T cell’s ability to target different TAAs, and it enhances the CAR-T cell’s capability to target biotinylated mediators recognising tumour antigens and not non-biotinylated ones [59]. The BBIR platform’s UCAR system can significantly improve CAR strategies and help to produce T cells with a potentially limitless range of antigen specificities. Preclinical studies on BBIR T cells have demonstrated the significant suppression of tumour growth in a xenograft mouse model of human ovarian cancer treated with a biotinylated antibody [75]. A similar UCAR has been generated by using the biotin–avidin setup, in which CD19+ and CD20+ cells were investigated as targets for anti-tag CAR-T cells [76]. However, a potential concern associated with those systems stems from the immunogenicity of avidin, which should be taken into consideration since the generation of specific antibodies may alter effector cells’ performance, trafficking, and persistence. Consequently, further investigation is necessary to evaluate the performance of the biotin–avidin anti-tag CAR-T cells in preclinical and clinical studies [58] to assess the real impact of those features. BBIR can efficiently recognise and bind a variety of biotinylated molecules. The principle behind this system is to introduce more than one switchable module that cooperates in different ways to generate logic circuitry [53].

### 2.5. Antibody-Dependent Cellular Cytotoxicity-Mediated Immune Receptor

The Antibody-Dependent Cellular Cytotoxicity (ADCC)-mediated immune receptor in UCARs consists of an intracellular Fc domain, which is connected to the extracellular adaptor domain composed of the Fc portion of a specific antibody (IgG). Therefore, the Fc receptor is attached to the tumour antigen via a bridge of specific antibodies linked to the antigen on the tumour cell surface. FcgRIIIa (CD16) is one of the most prominent examples of the Fc receptor connected to the Fc region of the IgG1 antibody, causing the activation of ADCC and tumour lysis. In ongoing clinical trials, the CD16 UCAR-T cell therapy has been investigated in HER2-positive cancer, in multiple myeloma (MM), and in non-Hodgkin’s lymphoma [65,89]. To delve into this, in the study by D’Aloia and colleagues, it has been shown that CD16A-CAR-engineered T cells can mediate granule-independent cytotoxicity and enhance anti-tumour efficacy by combining monoclonal antibodies (mAbs) with T cell immune responses [89]. Ochi et al. demonstrated that T cells engineered to express a chimeric CD16-CD3 receptor effectively mediate ADCC and proliferate and exhibit superior tumour-suppressive activity compared to NK cells, suggesting their potential as a more durable and effective alternative effector for monoclonal antibody cancer therapies [65].

### 2.6. The Bispecific Immune Receptor

The bispecific immune receptor of UCARs is engineered to simultaneously engage the tumour antigen and the extracellular domain of a universal receptor. Within this design, a bispecific antibody is linked to the various subsets of T cells, which is considered an advantage of UCARs compared to conventional CARs. The first bispecific immune receptor was designed by Urbanska et al. using the folate receptor a (FRa) as a genetically modified extracellular domain bound to the TCR intracellular domain. Accordingly, they showed that the bispecific antibody selectively redirected the UCAR-T cell toward the CD20+ tumour cells, targeting both CD20 and FRa simultaneously [77].

Despite high immunogenicity in humans, this concept opened the door to the modularisation of the CAR structure. The CAR indeed can be split into two parts: (i) the signalling module on T cells, consisting of the extracellular domain that specifically binds to the switching module and the intracellular domain that transmits the activation signals; (ii) the switching module, usually a bispecific antibody or small molecule, recognised by the signalling module on T cells that binds the antigen on cancer cells. This split, universal, and programmable (SUPRA) CAR approach, which will be discussed in more detail later on, has been exploited to recognise a variety of switching modules comprising neo-epitopes, SpyTag, biotin, FITC, and leucine zippers [87].

### 2.7. SUPRA CAR

To enhance the capabilities of CAR-T cells, researchers have developed the split, universal, and programmable (SUPRA) CAR system. SUPRA CARs can be finely controlled using various mechanisms to prevent excessive activation and respond intelligently to multiple antigens, enhancing their ability to target tumours with improved specificity. The SUPRA CAR is a construct composed of two components: (i) the universal receptor (zipCAR), which is expressed on T cells, and (ii) the tumour-targeting scFv adaptor (zipFv). The zipCAR is generated by combining intracellular signalling domains with a leucine zipper serving as the extracellular domain. The zipFv adaptor molecule is formed by fusing a complementary leucine zipper and a scFv that binds to the tumour antigen. The resulting complementary leucine zippers activate the zipCAR on T cells [57,58].

Unlike conventional CAR designs, SUPRA CARs are modular and allow for the targeting of multiple antigens without altering the patient’s immune cells genetically. To test its capability, previous studies reported that human primary CD8+ T cells engineered to express an RR zipCAR (RR leucine zipper motif), with different zipFvs designed to target various tumour antigens, demonstrated specific killing activity when co-cultured with tumour cells expressing the corresponding antigens. The split CAR design offers multiple adjustable variables that influence T cell response, including leucine zipper pair affinity, scFv-tumour antigen affinity, zipFv concentration, and zipCAR expression levels. For example, in previous studies, the concentration of zipFv required for T cell activation inversely correlated with leucine zipper pair affinity, and the expression level of zipCAR correlated with cytokine secretion and cellular activation. This split, universal, and programmable approach has been exploited to recognise other varieties of switching modules comprising neo-epitopes, SpyTag, biotin, and FITC [87]. FITC is a biocompatible fluorochrome employed in various biological and biomedical applications within physiological systems that can be easily conjugated to an Ab. This anti-FITC scFv can recognise the FITC-labelled monoclonal Ab, which is specific for a target tumour antigen. After the binding of the scFv to the FITC moiety, the intracellular domain of the T cell including CD28, CD3, and 4-1BB is activated (54). Anti-tumour mAbs labelled with FITC, such as anti-Her2 (trastuzumab), anti-CD20 (rituximab), and anti-EGFR (cetuximab), are fused to the anti-FITC CAR. This connection includes a specific linkage between the anti-FITC CAR and FITC-labelled Ab. This CAR-T cell has been reported to increase T cell proliferation, cytokine release, and tumour lysis. Accordingly, this strategy can be used to target different types of TAAs for cancer treatment [59].

Clinical trials of SUPRA CARs have been carried out for CD19/CD20 (NCT02776813) and CD123 (NCT04230265). Other targets under development include CD33, prostate stem cell antigen (PSCA), prostate-specific membrane antigen (PSMA), GD2, epidermal growth factor receptor (EGFR), cell-surface associated mucin 1 (MUC1), and sialyl-Tn (STn) [55]. Moreover, the CD123-specific targeting module has undergone additional deimmunisation steps to minimise potential immunogenicity, which has demonstrated both favourable tolerance and effective targeting within the human body [56]. The modular nature of the SUPRA CAR design allows for the precise tuning of T cell response and activation levels, which has the potential to address challenges related to toxicity often seen in high-affinity scFv CARs. Moreover, one well-known adverse effect of conventional CAR-T cells is CRS, arising from uncontrolled T cell responses. SUPRA CAR technology efficiently addresses the issue of CAR-T cell overactivation, which is achieved by employing a competitive zipFv, capable of binding to other zipFvs. By administering this competitive zipFv, the function of the zipCAR can be restrained when it is not required. Interestingly, Cho et al. [57] showed the application of multiple competitive zipFvs employed for the EE motif zipFv with a weak, medium, and strong affinity to demonstrate the effectiveness of this method. The leucine zipper of Cho et al.’s zipCAR contained the signalling domain CD3 as well as the costimulatory domains CD28 and 4-1BB. ZipFV’s scFvs were developed to specifically target the tumour antigens Her2, Axl, and mesothelin. In their study, the mouse xenograft breast cancer model was given both the traditional Her2-CAR and the SUPRA system, the -Her2-EE-zipFv-RR-zipCAR. Remarkably, the results showed that the SUPRA CAR system significantly reduced both tumour burden and antigen escape compared to conventional CARs. Results indicate that the SUPRA CAR functions as a highly specialised CAR-T cell that is intelligent, multi-targeted, and programmable. Overall, SUPRA CARs hold significant promise for clinical applications due to their capacity to effectively target multiple antigens while maintaining controllable activity [57,58].

### 2.8. Dual CAR

Dual CARs are composed of two distinct CAR architectures that have both primary (CD3-containing) and secondary (co-stimulatory) signalling domains. Each of them targets a different antigen on the surface of tumour cells. Furthermore, an increasing number of studies suggest that the simultaneous targeting of two tumour antigens can improve the specificity to tumour cells, reducing the chance of “on-target/off-tumour” toxicity and boosting CAR-T cells’ anti-tumour efficacy [9].

One study by Ruella et al. described an effective strategy for the treatment of B-ALL by targeting CD123 in combination with CD19. They demonstrated the feasibility of expressing two full CARs within a single T cell, showing enhanced efficacy in eliminating leukaemia cells, as compared to either CAR-T alone or a mixture of both CARTs, while overcoming potential antigen escape [90].

Another valid study on haematological malignancies focused on bispecific CD19-CD22-CAR-T cells, which were shown to specifically target CD19+ and CD22+ human leukemic cells, generate interferon (IFN)-, and efficiently eradicate the target cancer cells in immune-deficient mice. Moreover, adults with relapsed or resistant B cell acute lymphoblastic leukaemia demonstrated significant therapeutic benefits from this approach [49,91].

Researchers have also been investigating the feasibility of dual CAR-T cell therapy in solid tumours. Anurathapan et al. described the encouraging results of using modified dual CAR-T cells against PSCA and MUC1 in a pancreatic cancer xenograft model. Their research showed that immune cells were recruited, antigen immune escape was prevented, tumour cell death was significant, pro-inflammatory cytokines were produced, and on-target/off-tumour toxicity was decreased [92]. In the context of pancreatic cancer, Kloss et al. developed a combinatorial dual CAR approach, featuring first-generation CAR-T cells that provide partial activation upon binding to a prostate stem cell antigen (PSCA) via a CAR carrying only the CD3 domain (PSCA CAR). This system is paired with a chimeric co-stimulatory receptor (CCR) targeting a different antigen, such as prostate-specific membrane antigen (PSMA), which carries only co-stimulatory domains (PSMA CCR). The full activation of CAR-T cells is achieved only when both antigens are engaged, triggering both signal 1 (via PSCA CAR) and the co-stimulatory signal 2 (via PSMA CCR). They verified that only prostate cancers expressing both PSCA and PSMA antigens are destroyed by these modified T cells, while single-antigen-positive tumours are not [93], ensuring enhanced specificity and safety in CAR-T cell activation. Another study described the use of dual CAR cells against pancreatic cancer by developing an anti-carcinoembryonic antigen (CEA) CAR coupled to a mesothelin-specific CCR (MSLN-4/1BB) [43]. Dual CAR-T cells could successfully inhibit the growth of AsPC-1 tumour cells in xenografted mice [43]. In the haematological setting, a dual CAR approach has been developed to mitigate the on-target/off-tumour toxicities associated with targeting CD123 and CD33 in acute myeloid leukaemia (AML). This approach utilises an IL-3-zetakine combined with a CD33 co-stimulatory receptor, allowing for the safer and more selective targeting of AML cells [94].

Dual-targeted CAR-T cells have been shown to be effective against breast cancer too [95], for example, with CAR-T cells equipped with a split dual-antigen targeting system that required simultaneous activation upon engagement with ErbB2 and MUC1 TAAs. ErbB2-positive tumour cells that also expressed MUC1 were specifically targeted, stimulating an ErbB2-dependent immune activation with increased production of IFN-γ that synergistically promoted a cytotoxic response, although IL-2 secretion was relatively low compared to control CAR-T cells [95]. In conclusion, compared to other single-receptor-engineered cells, dual CAR-T cells show improved anti-tumour cytotoxicity when used against tumour cells expressing both antigens while not affecting normal cells.

### 2.9. Tandem CAR

Tandem CAR-T cells are one of the most relevant bispecific multi-targeted and programmable CAR technologies (TanCAR) consisting of a single receptor with two distinct antigen recognition domains. TanCAR technology allows T cells to target several tumour antigens simultaneously with good performance and safety. It is interesting to note that TanCAR and dual CAR vary, as TanCAR uses a single transmembrane glycoprotein with tandemly organised antigen-binding domains [96,97]. This alternative cytotoxic mechanism has been conceived to tackle tumour antigen escape, in particular, the loss of target antigens due to the CAR T selective pressure in a mono-targeting strategy. Moreover, Tandem CAR can be useful to address antigen heterogeneity [34]. Furthermore, TanCARs may reduce “on-target/off-tumour” harmful effects caused by uncontrollable classical CAR activity since they are designed to distinguish labelled cancer cells that express both target antigens but not healthy cells or tumour cells that only express a single positive antigen [96,98].

Hedge et al. developed a Tandem CAR-T cell to address glioblastoma in a xenograft model by employing HER2 and IL13R2 scFvs. They found out that the interaction between T cells and cancer cells was boosted by the simultaneous recognition of both tumour antigens. TanCARs’ cytotoxic-killing effect led to a higher production of cytokines such as IL-2 and IFN-γ, particularly against tumour cells showing both antigens compared to tumour cells that only expressed a single one [47].

Schneider et al., instead, designed a CD19-CD20-Tandem CAR and demonstrated its efficacy in leukaemia cell lines (in vitro and in vivo). In this study, they co-cultured TanCAR-T cells with leukaemia cell lines, in which the CD19 and CD20 antigens were simultaneously targeted. They managed to demonstrate effective cytotoxicity against tumour cells by significantly generating IL-2, IFN-γ, GM-CSF, and TNF-a with respect to non-engineered T cells. Additionally, CD19-CD20-TanCAR was more effective than a single CD19-CAR-T cell at eliminating cancer cells and reducing both tumour antigen escape and “on-target/off-tumour” toxicity [98]. In support of these results, the same evidence was later reported by the Zah et al. group regarding CD19-CD20-TanCAR [46].

Other TanCAR types have been developed, including the CD19-Her2-Tandem CAR and the CD19+/CD133+ TanCAR, to treat, respectively, mixed-lineage leukaemia and MLL. In both cases, TanCAR activity reduced “on-target/off-tumour” toxicity and CRS by targeting double-positive-antigen cells instead of cells expressing a single antigen [96,98,99].

### 2.10. SynNotch-CAR

Another promising, customisable, and multi-targeted CAR-T cell is the synthetic Notch receptor (SynNotch)-CAR, also known as the “AND-gate” CAR, which enables multimodal tumour antigen targeting [100]. SynNotch receptors are composed of several key components, including (i) synthetic external recognition domains (such as single-chain antibodies), (ii) a synthetic intracellular transcriptional domain, called the Notch intracellular domain (NICD), with a regulatory role in gene transcription, and (iii) the central core domain, which plays central role in the Notch signalling pathway and helps regulate the cleavage and activation of the SynNotch-CAR after binding to antigens [60,101]. The engagement with a specific antigen triggers a controlled transmembrane cleavage of the SynNotch receptor, mimicking a physiological Notch activation process. The intracellular transcriptional domain is released and transported to the nucleus, where it activates the expression of target genes under the control of the corresponding upstream cis-activating promoter [100]. Such an activation pathway differs from conventional CAR activation by scFv antigen recognition [102]. The SynNotch receptor is designed to produce responses in programmable pathways, in contrast to traditional CARs that lack any controlled mechanism. Moreover, because of the distinctive structure of the SynNotch receptor, it is possible to employ a synthetic transcription factor rather than the intracellular domain of Notch, and instead of the extracellular part, other substitutes can be introduced into the receptor structure [34].

Unlike conventional CARs, AND-gate T cells are designed to be activated only when two target antigens are present, while being inactive in the presence of a single antigen only. The subsequent expression of the CAR occurs following the activation of the SynNotch receptor. Since this controllable mechanism is not present in conventional CAR-T cells, the likelihood of side effects in the latter is increased compared to Syn-Notch CAR-expressing T cells. This class of CAR-T cells has the potential to produce a wide range of cytokines [60]. In addition, previous studies have reported the possibility of employing a synthetic Notch (synNotch) receptor system able to activate the expression of a secondary CAR upon binding to a specific antigen via the synNotch receptor [103,104].

Furthermore, T cells with multiple synNotch receptors serving as versatile regulatory adapters have recently been developed. These cells are capable of precisely targeting cancers by incorporating up to three distinct antigens while avoiding the recognition of tumours associated with only two antigens [61].

The synNotch-CAR has demonstrated superior tumour elimination compared to conventional CAR-T cells. This increased performance is achieved by preventing tonic signalling and exhaustion, and maintaining a larger proportion of T cells in a naive/stem cell memory state, ultimately resulting in more effective tumour eradication compared to traditional CAR-T cells [62,63].

The use of synNotch CARs has been reported and described by several authors. Choe et al. tested a SynNotch-CAR to target specific antigens for glioblastoma, such as the variable but tumour-specific glioblastoma neoantigen epidermal growth factor receptor variant III (EGFRvIII), the tissue-specific antigen for the central nervous system (CNS), and the myelin oligodendrocyte glycoprotein (MOG). Administering a single intravenous infusion of EGFRvIII synNotch-CAR-T cells in immunodeficient mice bearing intracerebral patient-derived xenografts (PDXs) with a heterogeneous expression of EGFRvIII resulted in higher anti-tumour efficacy, improved T cell survival, more stem-like phenotypes, less exhaustion, and enhanced in vivo permanence compared to traditional CARs [63]. In a second study, Hyrenius-Wittsten et al. detected the tumour-specific antigen Alkaline Phosphatase Placental-like 2 (ALPPL2) as a specific target expressed in a variety of solid malignancies, including mesothelioma and ovarian cancer. By testing the activity of SynNotch CAR-T cells in mice models of human mesothelioma and ovarian cancer, they observed a greater exhibition in controlling the tumour volume when compared to conventional CARs [42]. Additionally, the SynNotch-controlled expression helped to prevent CAR-mediated tonic signalling, allowing T cells to preserve a long-lasting memory and non-exhausted state [64].

### 2.11. TRUCKs

Treating solid cancers with CAR-T cells has proven significantly less successful compared to the outcomes obtained in haematological malignancies, often resulting in transient and poor remissions. This is attributed to T cell hypo-function and exhaustion within the immune-suppressive TME. T cells from non-responsive patients retained inhibitory receptors like PD-1 and CTLA-4, leading to the reduced production of pro-inflammatory cytokines and limited proliferation. This functional exhaustion, often caused by continuous antigen stimulation, prevents most CAR-T cells from penetrating the tumour, leaving them at the tumour margin and unable to exert their effects within the tumour. Therefore, strategies aimed at preserving the functional cytotoxicity of T cells and improving their trafficking inside the tumour mass are essential and of the utmost importance for the success of CAR-T cell therapy. Recent clinical results have indeed highlighted that non-responsive patients accumulate T cells with an early memory and exhaustion signature after CAR T cell therapy, in contrast to responders [64]. The fourth generation of CAR-T cells, known as TRUCKs, are genetically engineered to release cytokines and other biological substances. TRUCKs have been developed to tackle a variety of malignancies, including glioblastoma, breast cancer, and other solid tumours, employing several cytokines such as IL-2, IL-12, IL-15, and IL-18 [9,82]. For instance, sustaining the function of cytolytic CD8+ T cells relies on a network of transcription factors, with FoxO1 playing a key role. The balance between FoxO1 and T-bet transcription factors affects T cell function. By inducing FoxO1-low and T-bet-high T cells, the cytolytic CD8+ T cell attack can be maintained, offering prolonged immune responses to advanced tumours. IL-18 has been identified as a potent cytokine capable of inducing this desired T cell phenotype. These observations have been supported by recent findings in which CAR T cell activity was enhanced in the presence of IL-18, with IL-18 released by engineered CAR-T cells promoting acute inflammation and reducing exhausted cell numbers within the tumour, resulting in a more effective immune response against established tumours [64].

In a study by Krenciute et al., IL-15 expression and secretion in IL13R2-CAR-T cells showed the enhanced cytotoxic activity of transduced T cells towards glioma tumour cells. Previous investigations have also reported that IL-12 induces immune-boosting properties in addition to IL-15, which can be achieved by reconfiguring suppressor cells in the TME and stimulating the innate immune system’s anticancer defences, including natural killer (NK) cells [105]. Moreover, TRUCK T cells increase the anti-tumour effectiveness of CAR-T cells by destroying target cells that are inaccessible to the CAR-T cells and by boosting the levels of IFN-γ, IL-6, and IL-27 cytokines. TRUCK T cells’ expression of IL-12 can also modify the tumour stroma in order to enhance its suitability for the function and persistence of the engineered T cells [82,106].

### 2.12. Physiological CAR

A physiological CAR is a programmable CAR structure that relies on a ligand–receptor interaction normally occurring in vivo. Instead of an scFv linked to the CD3 intracellular signalling domain, a ligand or receptor is adopted as an extracellular antigen-binding domain. The antigen-binding domain of the receptor specifically targets a ligand expressed on tumour cells. Compared to conventional CARs, these CAR-T cells can directly induce T cell activation, and have strong cytotoxicity and less immunogenicity [68,69].

Different studies have reported the use of physiological CARs. Zhang et al. targeted the B7-H6 ligand on cancer cells by using the NKp30 receptor. The findings showed that IFN-γ was produced at high levels and that killing activity was significantly cytotoxic and selective [70]. Shaffer et al. developed a physiological CAR able to target CD70 (the receptor’s ligand on tumour cells) by employing the whole length of CD27 as a receptor. The direct activation of T cells resulted in the efficient clearance of CD70+ cancer cells through ligand–receptor linkage, and T cells were also protected from activation-induced cell death (AICD) [107]. Other studies on glioblastoma testing a physiological CAR expressing the IL13-zetakine targeting IL13R2+ tumour cells showed a strong cytolytic function and a significant quantity of inflammatory cytokines produced [71]. Zhang et al., employing the natural killer group 2 member D (NKG2D) receptor, created another physiological CAR that recognises the human NKG2D ligand. According to their research, T cells were directly activated, diverse human NKG2DL+ tumour cells were effectively eradicated, inflammatory cytokines were produced, and the progression of lymphoma cells was inhibited [108]. Additionally, a VEGFR-specific physiological CAR, developed to target malignant cells expressing VEGFR, was reported to successfully eliminate VEGFR+ tumour cells [66]. In another investigation by Muniappan et al., HER3 and HER4 receptors were used against heregulin in the evaluation of manufactured physiological CAR, which indicated potent cytotoxic effects on breast cancer cell lines [109].

Walseng et al. used a TCR CAR that included cysteine to improve TCR dimer stability in T and NK cells by abandoning the conventional CAR design [67]. Similar outcomes were observed by Liu et al. by designing a chimeric receptor using an immunoglobulin-heavy chain fused to TCR-Cα and an immunoglobulin-light chain fused to TCR-Cβ. They demonstrated that the tonic signalling that presents difficulties for some CAR designs is absent from this double-chain design that mimics the TCR architecture. Moreover, compared to conventional CAR designs, the TCR-like approach demonstrated better specificity against targeted antigens [72]. Given these qualities, physiological CAR-expressing T cells promise a more effective CAR-T cell treatment for cancer as compared to the traditional CAR-T cells [34].

## 3. Modification on Non-CAR Molecules

Although extensive attempts have been performed in order to modify the CAR structure with the final goal to improve transduced effector cell functions, particularly against solid tumours, these approaches bounce against the hurdles imposed by the TME. Indeed, the TME prevents T cells from penetrating the tumour mass, exacerbates effector cell exhaustion, and harms their survival [110].

In fact, the TME deploys a series of factors that hinder the accessibility and efficacy of CAR-Ts. TME suppresses the infiltration of T cells by reactive oxygen species (ROS) production from myeloid-derived suppressor cells (MDSCs) [111]. ROS accumulation contrasts T cell anti-tumour function by altering T cell receptors (TCRs) and CD8, downregulating CD3ζ expression, and impairing the antigen-specific response [112]. MDSCs in tumours release arginase, depriving T cells of crucial amino acids and lowering IL-2 and IFN-γ levels [113,114]. Excess ROS from environmental stressors promotes mutations and tumorigenesis [115]. Additionally, tumour cells’ high metabolic demands starve T cells of nutrients, diminishing CAR-T effectiveness. Metabolites like lactate, adenosine, and arginase-1 further inhibit T cell activity [116,117,118,119]. MDSC-derived methylglyoxal (MG) disrupts CD8+ T cells, linking to oxidative stress and the NRF2 pathway [120,121]. Overall, maintaining equilibrium between T cell and tumour metabolism is essential for CAR-T therapy to be effective [122].

Other limits are represented by the presence of inhibitory ligands such as PD-L1 on cancer cells, which might decrease T cell activity and efficacy, as well as antigen heterogeneity and tumour antigen escape [123]. Such a complex scenario with so many challenges for CAR-T cell functioning has prompted the development of integrated strategies with CAR cells.

### 3.1. Impact of Cytokine Production on CAR-T Cells

Pre-clinical and clinical studies demonstrated that arming CAR-T cells with cytokines enhanced their anti-tumour responses. Coupling CAR-T cells and cytokines generates a synergistic impact that has shown significant promise for cancer treatments. For example, Kloss et al. described that PSMA-directed CAR-T cells can be significantly potentiated by the production of cytokines that also improve T cell growth, long-term in vivo retention, and resilience to exhaustion [124]. Cytokines are pleiotropic hormones and comprise ILs, TNFs, IFNs, chemokines, colony-stimulating factors (CSFs), and growth factors. They play a variety of functions in immunity, including the stimulation, growth, specialisation, and translocation of immune cells [125]. Furthermore, CAR T cell-derived IFNγ may play a vital role in enhancing CAR T therapy for solid tumours by influencing the TME in multiple ways. Specifically, IFNγ activates macrophages [126] and microglia [127], recruits and stimulates cytotoxic T cells, polarises CD4+ T cells into Th1 effector cells, and inhibits tumour-promoting regulatory T cells (Tregs) [128,129]. This multifaceted immune activation could be crucial for advancing CAR T therapies for solid tumours, such as glioblastoma (GBM), where clinical responses have been limited compared to haematological malignancies, but early studies show safety and initial bioactivity in selected cases [130,131,132,133]. For instance, an IL13Ra2-CAR targeting a GBM-associated antigen has shown promising results in trials, including one patient who achieved a complete response [134]. This clinical experience suggests that CAR-T cells not only target antigen-positive cells, but also influence the TME by driving localised changes in inflammatory cytokines, reshaping immune cell infiltrates, and affecting clonal T cell populations. As a second example, Hongping and colleagues developed a CAR-T cell therapy targeting transmembrane tumour necrosis factor-alpha (tmTNF-α), which is overexpressed in certain breast cancers, including triple-negative breast cancer (TNBC). These tmTNF-α-specific CAR-T cells demonstrated strong cytotoxicity against tmTNF-α-positive breast cancer cells in vitro and in vivo, accompanied by the increased secretion of IFN-γ and IL-2. In TNBC-bearing mice, tmTNF-α CAR-T therapy led to significant tumour regression, improved survival, and elevated serum IFN-γ and IL-2 levels. However, tmTNF-α was also found to induce PD-L1 expression via the p38, NF-κB, and AKT pathways, potentially limiting CAR-T efficacy. Blocking the PD-1/PD-L1 pathway with anti-PD-1 antibodies significantly enhanced the anti-tumour effect of tmTNF-α CAR-T therapy. This combination proved effective against primary tumours and showed promise for metastatic control, indicating a potential therapeutic approach for tmTNF-α-positive breast cancers, including TNBC [135].

Other examples of combination therapies with cytokines for solid tumours come from the use of recombinant IL-12. At least three different observations explain how IL-12 contributes to tumour regression: CAR-T cell survival and persistence improvement, shifting inflammation status from chronic to acute that is conducive to immunity in the TME, and both attracting and stimulating innate immune cells to trigger further anti-tumour responses [136,137,138]. Regarding this aspect, Yeku et al. reported that, in the inhibitory TME of murine ovarian peritoneal carcinomatosis, armoured CAR-T cells that were able to secrete IL-12 increased survival against ovarian tumours [137]. Adachi et al. modified CAR-T cells to express both IL-7 and CCL19, and they observed complete remission in mice harbouring solid tumours as compared to standard CAR T therapy [139]. Similarly, Luo et al. engineered CAR-T cells to produce IL-7 and CCL21 showing a greater therapeutic potential versus solid tumours than standard CAR therapy [140]. Additionally, it has been observed that CAR-T cells grown with IL-15 can sustain a less differentiated stem cell memory phenotype with decreased exhaustion and increased proliferation following target engagement [141,142]. A variety of cell types, such as macrophages and dendritic cells, can release IL-15, which can drive CD8+ T cells and NK cells to proliferate faster and be more effective against tumours [143]. Comparing GD2-targeting CAR-T cells to conventional CAR-T cells, the incorporation of IL-15 production has shown stronger anticancer activity both in vitro and in vivo [144].

### 3.2. Chemokine Receptors as “Attractive” Factors for CAR-T Cells

Another important limit hindering immunotherapy effectiveness for solid tumours is inadequate infiltration inside the tumour mass and trafficking. To overcome these shortcomings, CAR-T cells expressing specific chemokine receptors have been designed to improve tumour targeting, enhance trafficking, and exploit the chemokines released by the tumours. For example, the chemokines CCL17 and CCL22 released by Hodgkin lymphoma attract regulatory T and T helper 2 cells as they express the C-C chemokine receptor (CCR) type 4. However, CD8+ T cells, which do not express CCR4, are not attracted to the tumour. Previous studies have indeed reported that intravenously administering CCR4-expressing CAR-T cells to mice with human Hodgkin lymphoma results in enhanced tumour homing and increased anticancer activity [145].

Moreover, IL-8 has also been found to function as a chemoattractant factor for neutrophils and other suppressive cells, thereby enhancing the immunosuppressive features of TME. Interestingly, CAR-T cells engineered to express the C-X-C motif chemokine receptor (CXCR) type 2, a receptor for IL-8, showed increased homing and anti-tumour efficacy in several solid tumour models in vivo, suggesting that IL-8 secretion can be exploited to achieve better treatments [142,146,147]. The co-expression of other types of chemokine receptors, such as CCR2b that enhanced transmigration and cytolytic activity of transduced T cells in vitro [148], CXCR3 [149,150], CXCR4 [151], and CCR6 [152], have also been tested with encouraging results.

### 3.3. Heparanase Production by CAR-T Cells

The extracellular matrix (ECM), a key element of tumours, represents a source of multiple communicating substances, provides mechanical support, and contributes to the regulation of the TME. Alongside these supportive functions, the ECM also presents a challenge for T cell infiltration. The rigidity of the ECM is frequently caused by the interaction between cancer cells and the TME during carcinogenesis, which promotes abnormal mechanotransduction and subsequent tumour progression [153]. Endogenous T cells may secrete specific enzymes capable of degrading the ECM of solid tumours. However, when T cells are engineered to generate CAR-T, they may experience a loss in the ability to degrade the ECM. In fact, the release of specific enzymes, such as heparanase (HPSE), by T cells is essential for ECM degradation [154,155]. HPSE, produced by activated T cells, exists as a latent precursor before becoming enzymatically active [156]. Ex vivo studies revealed that long-term-expanded T cells (LTE-T) lose HPSE expression and enzymatic activity, hindering their ECM degradation capacity, compared to freshly isolated and briefly activated T cells [157,158]. This loss may stem from the accumulation of full-length p53 protein, which binds to the HPSE gene promoter, potentially impairing T cell migration and anti-tumour effects in adoptive immunotherapy [159,160].

While other enzymes like metalloproteases can modify ECM components, some are also downregulated upon TCR activation and cytokine exposure. To enhance the ECM degradation capacity of CAR-expressing LTE-T, Caruana et al. [161] proposed inducing HPSE expression without affecting the CAR viability, expansion, or function. This strategy could potentially improve the anti-tumour activity of CAR-redirected T cells in patients with solid tumours rich in stroma [162].

Particularly, engineering LTE-T cells to express HPSE through retroviral gene transfer improved ECM degradation and invasion capacity. HPSE-expressing LTE-T cells paired with a GD2-specific CAR showed strong anti-tumour activity against neuroblastoma (NB) cells, especially in ECM-rich environments. Functional assays confirmed ECM degradation specificity, with HPSE-inhibitor blocking invasion. CAR(I)HPSE+ LTE-T cells also maintained cytolytic activity and Th1 cytokine release comparable to CAR+ cells, without increasing activation-induced cell death. These results support HPSE-CAR co-expression as an effective strategy to enhance LTE-T efficacy in ECM-rich tumours [161].

### 3.4. Dominant-Negative Receptors

Solid tumours are characterised by extensive local immunosuppression; therefore, strategies aimed to fight the immunosuppressive cues in CAR-T cells are extremely desired and investigated.

Furthermore, CAR-T cells need to bypass immunosuppressive pathways such as PD-1 and CTLA-4 to be effective in solid tumours. In prostate cancer, for instance, TGF-β creates an immunosuppressive environment, promoting metastasis and suppressing immune response [163]. Blocking TGF-β signalling with a truncated receptor, dnTGF-βRII, has been shown to enhance anti-tumour immunity, increasing T cell infiltration and activity within tumours [134]. Studies in prostate cancer mouse models confirm that dnTGF-βRII expression can lead to strong anti-tumour responses and improved T cell infiltration [124,164]. However, PSMA-targeted CAR T cells, despite their specificity, have demonstrated limited persistence and response, revealing a need for more robust CAR constructs [165]. Kloss et al., then, thought to combine dnTGF-βRII with a PSMA-specific CAR to boost T cell infiltration, persistence, and efficacy, offering a promising therapeutic advance for prostate cancer treatment [124]. Their findings showed that PSMA-specific CAR-T cells engineered to be insensitive to TGF-β signalling (dnTGF-βRII-T2A-Pbbz) can selectively target and eradicate advanced PSMA-expressing tumours. These modified CAR-T cells also show a significant proliferative advantage over conventional PSMA CAR-T cells that remain responsive to TGF-β. This has meaningful clinical implications, as solid tumour CAR-T cell trials have generally reported poor CAR T cell expansion [166]. It is also possible to inhibit PD-1 pathway, thanks to the co-expression of PD-1 dominant-negative receptors. PD-1 is a receptor that normally shows low expression on healthy T cells, but is upregulated on exhausted T cells in cancer patients [167]. PD-L1, the ligand for PD-1, is commonly present on various cancer cell surfaces. The PD-1/PD-L1 interaction contributes to T cell apoptosis, anergy, and exhaustion, weakening immune responses. Blocking PD-1 or PD-L1 can partially restore T cell function, and several studies have shown that PD-1-targeted monoclonal antibodies enhance CAR-T cell anti-tumour activity [168,169,170,171]. Beyond antibodies, PD-1 gene editing has also been shown to boost CAR T cell efficacy [171].

In this regard, Guodi et al. investigated PD-1′s role in T cell exhaustion using mesothelin (MSLN)-expressing ovarian (SKOV3) and colon cancer (HCT116) cell lines. By employing an shRNA-mediated PD-1-silencing approach intrinsic to CAR T cells, they achieved a significant reduction in PD-1 on the T cell surface, which in turn significantly enhanced CAR T cell cytokine production and cytotoxicity against PD-L1-expressing tumour cells in vitro. This approach demonstrated the improved anti-tumour efficacy of PD-1-silenced, MSLN-targeted CAR-T cells across multiple cancer models and underscores the therapeutic potential of targeting immune checkpoints like PD-1 to boost CAR T cell therapies against various tumours [172,173].

To mention another relevant study, Kalinin et. al. used VHH domains from anti-PD-1 nanobodies to block PD-L1 binding and induce PD-1 loss on CD19 CAR-T cells [174]. In fact, in CD19 CAR-T therapy for B cell malignancies, the upregulation of inhibitory receptors like PD-1 and CTLA-4 limits T cell activation and efficacy. Elevated PD-L1 levels on tumour cells induce PD-1 expression on CAR-T cells, leading to negative signalling that hinders T cell stimulation and cytotoxicity. Co-expressed PD-1, CD19 CAR, and a PD-1-targeting VHH fused with the PEBL motif, which is known to relocate proteins to the Golgi/ER. The authors additionally observed reduced CAR-T cell survival and rapid differentiation, aligning with other findings that PD-1 signalling is crucial for CAR-T cell longevity [174]. Although short-term assays showed comparable cytotoxicity, sequential killing experiments indicated that the absence of PD-1 signalling negatively impacts CAR-T cell persistence and efficacy. This important work is the first example demonstrating that the VHH domain of the 102c3 nanobody effectively blocks PD-1/PD-L1 interactions. While this approach increases CAR signalling, it adversely affects CAR-T cell survival and functionality. Therefore, alternative immune checkpoint strategies should be considered to enhance CAR-T persistence and effectiveness in the TME.

### 3.5. CAT-CARs

The insufficient and ineffective vascularisation of the tumour mass locally exacerbates hypoxia that is often a hallmark of a TME, and is characterised by a high density of closely packed tumour cells and a dense ECM with structural and physiological abnormalities [175,176]. Hypoxic conditions impact on vascularisation through a variety of processes comprising the activation of hypoxia-inducible factors (HIFs) [177]. HIFs promote tumour development, infiltration, metastasis, and angiogenesis [178]. Numerous solid cancers overexpress HIF-1α in response to hypoxia [179], and this condition promotes cancer spreading and resistance to CAR T treatment since T cells require oxygen for their growth and functionality. Additionally, the Warburg effect boosts TME acidity through the production of lactate that damages the T cell’s functioning. Reactive oxygen species (ROS) are produced, causing oxidative stress and hindering the effectiveness of CAR-T cells in these conditions. To overcome these barriers, researches developed CAT-CARs, where CAT stands for T cells engineered to release catalase, an antioxidant enzyme that metabolises and degrades H2O2 in the TME. By lowering the oxidative stress state inside the TME, CAT-CARs decrease ROS concentration [176,180].

Employing antioxidant agents could be another strategy to restrain T cell death, induced by tumour metabolic stress. To this aim, the Scheffel research group showed a significant reduction in T cell death by using N-acetyl cysteine (NAC) as an antioxidant agent in combination. This finding holds promise in fighting the oxidative stress induced by the TME, potentially paving the way to the development of CAR-T cells capable of secreting antioxidant factors or to synergistic strategies combining antioxidants with CAR-T cell therapy [181].

Another approach proposed to physically restrain and boost CAR T cell activity in hypoxic conditions is based on the design of a CAR endowed with an oxygen-sensitive subdomain from HIF1a transcription factor. This CAR can stimulate T cell activation only in the hypoxic TME, while preventing stimulation in healthy and well-oxygenated tissues. Undoubtedly, this strategy offers the potential for the more precise targeting of tumours, leading to the development of self-decision-making CARs that activate only in the target sites, thereby sparing normal sites and reducing possible side effects [182].

### 3.6. Genome Editing of CAR-T Cells

Genetic manipulation approaches can eliminate certain DNA regions within the T cell genome that are implicated in aberrant and uncontrollable immune responses or immunosuppressive activities elicited by the tumour. One approach takes advantage of synthetic nucleases that produce sequence-specific targeted DNA double-strand breaks (DSBs). These nucleases contain a non-specific cleavage domain that initiates DNA double-strand breaks and a sequence-specific DNA-binding domain. Two major cellular DNA repair pathways, homologous recombination (HR), and non-homologous end joining (NHEJ) can be triggered in response to DNA breakage to restore DNA integrity [183]. ZFN, TALEN, and the bacterial nuclease CAS9 (CRISPR/CAS9 system) are enzymes that have emerged to genetically modify the genome in a sequence-specific manner. A restriction enzyme known as Fok1 provides the nuclease domain for the recombinant protein ZFN [58,78]. Through the use of a linker domain, this domain is linked to a binding domain made up of three to four zinc fingers that can bind particular DNA sequences. The ZFN is intended to function as a specific endonuclease of DNA that can cause a DBS break in a user-defined genomic area. In contrast, the nucleasic domain only functions as a dimer. Considering that each zinc finger can only recognise a 3–4 bp sequence on DNA, the complete enzyme is highly selective for an 18–36 bp sequence.

Genome editing techniques impinging into the NHEJ machinery have been applied to target and prevent the HLA-A and the TCRα constant region (TRAC) genes [51,78]. Actually, large nucleotide segments may be deleted or inserted during the cell cycle as a result of the NHEJ pathway and homology-directed DNA machinery repairing DNA double-strand breaks. This can have an impact on the expression of particular target genes [184,185]. Numerous organisms [79,186,187,188] employ methods based on the site-specific endonuclease TALEN to complete gene-specific genome editing. The cleavage activity of TALEN is much more specific than that of ZFN because it possesses an additional DNA-binding domain with a highly conserved amino acid sequence of 33–34 residues. For instance, the TCRα (TRAC) constant region can be selectively gene-edited using TALEN-based technology to disrupt the gene and reduce T cell activation against allogeneic antigens linked to GVHD. This lowers the likelihood that CAR-T cells will successfully engraft [79]. CRISPR/Cas9 offers even greater flexibility, manoeuvrability, and relative accuracy compared to TALEN, opening the possibility of multiple gene editing. Currently, CRISPR/Cas9 has been employed in several clinical trials, including UCART019 targeting CD19 (NCT03166878), CTX130 targeting CD70 (NCT04502446, NCT04438083), CTX120 targeting BCMA (NCT04244656), and CT125a targeting CD5 (NCT04767308). For the expression of CAR, CD19-specific CAR is knocked into the TRAC locus of T cells, by which its expression is enhanced and unified under the control of the TCR promoter [189,190]. In UCART7 targeting CD7 for T cell malignancies, TRAC and CD7 are simultaneously knocked down, the former for preventing GVHD and the latter for preventing the fratricide of the very effector cells [191].

## 4. Stimuli-Inducible Approaches

### 4.1. Small-Molecule-Controllable CAR-T Cells

Alternative methods for regulating the dosage and activation of T cells, and for reducing adverse effects like CRS, include chemically controlled CARs. Examples of these controllable and reversible approaches are the ON-switch CAR, like the split-CD19CAR, in which the formation of functional CAR molecules only occurs after the administration of the small molecule rapamycin analogue AP21967 (rapalog) [192]. Other ON-switch systems regard the EGFRvIII CAR [193] or the tetracycline (Tet)-On system (an inducible gene expression mechanism for mammals), controlled by the administration of the small molecule doxycycline (Dox) [194]. Another method concerns inducible CAR degradation in T cells employing the ligand-induced degradation (LID) domain, such as degron and the dihydrofolate reductase (DHFR) destabilising domain [195,196]. The CAR-LID fusion protein is more readily degraded by proteasomes when a small molecule ligand for degron-shield-1 is present ([195,196]). Contrarily, trimethoprim (TMP) can stabilise CAR-DHFR, allowing for the drug-dependent regulation of CAR expression and activity [196]. Another method to module CAR-T cells is by using the proteolysis-targeting chimaera (PROTAC) that can interact with E3 ubiquitin ligase and target protein ligands, which cause the target protein to be degraded by the ubiquitin–proteasome system [197]. After the inclusion of PROTACs, the E3 ligases can break the CD19 CAR molecule connected to the bromodomain from the BRD4 protein; nevertheless, the PROTAC approach holds some disadvantages like the possible destruction of endogenous proteins in addition to the CAR protein, which may be harmful to CAR-T cells as well [197]. Further, Yang et al. thought to exploit an inducible gene expression system to achieve CAR-T cell modulation by employing the compound resveratrol (RES) and showed that RES-activated CAR-T cells exhibited similar in vivo cytotoxic effects compared to standard CAR-T therapy with the advantage that RES-based inducible gene expression systems can serve as OFF switches [198].

Another strategy is to introduce suicide systems, such as transgenic enzymes like the herpes simplex virus–thymidine kinase (HSV-TK), which can trigger acute cytotoxicity through the pro-drug (Ganciclovir, GCV) [199], or the apoptotic gene caspase-9 (iCasp9), which can be activated through the biomolecule AP1903 (Rimiducid), which induces the dimerisation of iC9 and activation of the apoptosome [200]. The irreversibility of this strategy has the downside that CAR-T cells are killed and not available any more [201]. In addition to inducible cell suicide strategies, “inactivating” approaches may be employed. Dasatinib, an FDA-approved tyrosine kinase inhibitor, has been demonstrated to temporarily inactivate CAR-T cells, and to improve the in vivo anti-tumour effect by lowering acute toxicity, inhibiting tonic CAR signalling, and preventing the functional exhaustion of T cells in vivo [196,202,203]. Decitabine, a DNA methyltransferase, may enhance the persistence, cytokine secretion, memory, and activity of CAR-T therapy by epigenome reprogramming. Since these compounds are safe and FDA-approved, their use as an on/off switch in CAR-expressing cells should be simple and straightforward [204].

### 4.2. CAR-T Cells Controlled by Light

Optogenetics is the ability to visually control biological processes with high spatiotemporal precision in target cells both in vitro and in vivo [205,206]. More precisely, controlling processes are exerted through the creation of genetically encoded light-sensitive proteins [207]. In a study by Kennedy et al., it was demonstrated how blue-light-controllable dimers, such as cryptochrome 2 (CRY2) and CIB1, can be used for controlling protein transcription, translocation, and pre-mediated DNA recombination. This study showcases the application of a new generation of optogenetic tools for remotely monitoring and manipulating cellular processes [207,208]. Further, Huang et al. developed the LINTAD (light-inducible nuclear translocation and dimerisation) gene activation system (biLINuS) and, by using this technology, it was possible to control CAR expression in T cells and eliminate tumours in vivo under the control of blue light [209]. To cite other examples, Zhang et al. (as described above), by adding a photocleavable linker between the FITC and folate moieties in the bispecific adaptor, designed a photoswitchable CAR to permit the deactivation of CAR-T cells by light [210]. Both O’Donoghue’s and He’s research groups further engineered optoCAR by achieving photoinducible anti-tumour activity on target cells. It is clear that light provides fine temporal and spatial control; however, its use in patients at this point may be not applicable for deep tumours due to its weak tissue penetration [211,212]. To overcome this barrier, other technologies have been considered such as near-infrared (NIR) light optical fibres, upconverting nanoparticles, and implanted LEDs [213,214,215]. For instance, Nguyen et al. showed anticancer responses of light-switchable CAR (LiCAR) T cells stimulated by NIR light using upconversion nanoplates [216]. In general, this optogenetics application in CAR-T cell therapy may increase their controllability and effectiveness [209]. However, the clinical usefulness of optogenetics in the CAR T field will need to be further assessed in the future.

### 4.3. Ultrasound or Heat to Control CAR-T Therapy

Ultrasound allows for the monitoring of internal organs and their physiology. Although ultrasound is a frequently used diagnostic imaging procedure, it has a restricted ability to monitor and control cellular processes. Thanks to the advent of biomolecular tools, recent developments have started to solve this issue by enabling ultrasound to interact directly with cellular processes like gene expression [217]. For example, Pan et al. used this approach with CAR-T cells, demonstrating the in vitro death of tumour cells induced by ultrasound by mechanically disturbing microbubble-coupled cells. Through this mechanical perturbation, they managed to activate the mechanosensitive ion channel Piezo1, which led to subsequent molecular events, including calcium influx, NFAT translation, and NFAT-mediated gene expression [218]. Another approach used in clinics to remove malignancies is Focused Ultrasound (FUS), which can raise the temperature in a small and restricted area. For instance, ultrasound can also result in localised hyperthermia because internal friction transforms mechanical energy into thermal energy. Scientists have exploited FUS to activate Hsp-driven transgene expressions by inducing mild hyperthermia (in vitro and in vivo), which is inspired by the endogenous heat shock response: heat triggers the heat shock promoter (Hsp), driving the expression of heat shock proteins [219,220,221,222]. An FUS CAR-T (with an Hsp-driven Cre-lox switch that FUS can monitor) was created by Wu et al., and in two subcutaneous tumour models this FUS-activated FUS-CAR-T cell showed specific anticancer efficacies. Interestingly, these types of cells were proven to induce much less on-target/off-tumour toxicity than conventional CAR-T cells [223]. Another strategy was conducted by Miller et al. by activating CAR-T cells with Hsp-driven IL15 superagonist using plasmic gold nanorods to convert NIR light to heat. According to their findings, NIR-activated IL-15 production increased CAR-T cells’ in vivo anticancer efficacy [224].

### 4.4. Virus-Mimetic Fusogenic Nanovesicles, Velocity Receptors, and In Situ Reprogramming of T Cells In Vivo

For this last paragraph, we decided to report on recent innovations in in vivo CAR-T therapy generation.

The first example comes from the work of Zhao G. and colleagues, where CAR-T cells are produced in vivo by using virus-mimetic fusogenic nanovesicles (FuNVs) by membrane-fusion-mediated CAR protein delivery. The authors created two T cell fusogens, p14TF and MVTF, by incorporating αCD3 scFv into the fusogens of the measles virus and reovirus, respectively, to facilitate the fusion of FuNVs with the T cell membrane. Compared to standard CAR-T cell treatment, this approach has the advantage of accelerating and simplifying the procedure. Moreover, the temporary introduction of the CAR protein should lower the likelihood of insertional mutagenesis and transformation [225]. Another innovative proposal comes from the work of Adrian C. Johnston and colleagues, who specifically focused on the urgency of solving critical challenges associated to CAR-T therapy for solid tumours. They observed that T cells exhibit different characteristics related to migration at low and high densities. A paracrine pathway of cytokines (IL5, TNFα, IFNγ, and IL8) promotes the highly migratory phenotype. They aim to take advantage of this discovery to create and express receptors that they name velocity receptors (VRs) by trying to stabilise this migratory pathway. By using these cytokines, VRs boost T cell motility and infiltrate in various solid tumours (lung, ovarian, pancreatic) in quantities that limit the ability of CAR-T cells to spread and disperse outside the tumour. They assessed that VR-CAR-T cells greatly reduce tumour development and increase overall survival compared to CAR monotherapy [226].

### 4.5. Hypoxia-Responsive CAR-T Cells

Another interesting example of innovation in terms of alternative CAR-T cell engineering comes from the work of Zhu X. et al. In contrast to recently published hypoxia-sensing techniques employed in CAR-T cells, they present in their study the development of a hypoxia-responsive CAR-T (5H1P-CEA CAR-T). Their CAR design increases CAR-T cell activation selectively in the hypoxic tumour environment while keeping the cells in a “resting” state during manufacturing. As a result, CAR-T cells are boosted in their anti-tumour activity against both tumour-cell-line-derived xenografts (CDX) and patient-derived xenografts (PDX). Upon tumour stimuli, the 5H1P-CEA CAR-T cells showed less exhaustion and had improved immunophenotypic features throughout manufacturing. Furthermore, a correlation was seen between CAR-T cells’ activation state and CAR expression in 5H1P-CEA CAR-T cells. In resting cells, the CAR is expressed at a low level, while, when cells are stimulated, for example, by a hypoxic environment, CAR expression is boosted in 5H1P-CEA CAR-T cells [227]. Additionally, we also want to mention the in situ reprogramming of immune therapies with the help of synthetic nanomaterials, which was discussed in a brilliant overview by Nie S. and colleagues. Thanks to this approach, the major issues with cell therapies may be resolved: the materials employed in in situ conversion are easily accessible in settings that are more convenient for patients as they consist of off-the-shelf drugs that can be produced in large quantities with less complexity and a lower cost. Furthermore, the in situ method can significantly conserve cell survival since it does not require cell separation, allowing the targeted cells to remain in their own physiological environment [228]. In particular, synthetic nanomaterials such as lipid nanoparticles LNPs and polymers promise alternatives to viral vectors due to their low immunogenicity, scalability, and cost-effectiveness. Moreover, they can be engineered with various ligands to enhance the targeting of specific cell types, which indicates that these synthetic nanomaterials enable effective transgene delivery for the in situ reprogramming of T cells, simplifying CAR-T cell production and reducing immunogenicity and side effects associated with traditional therapies. Startups like Umoja Biopharma and EXUMA Biotech are developing these in situ CAR-T therapies using both viral and nonviral methods. The initial preclinical results for synthetic nanomaterial-based approaches are promising [229,230].

## 5. Conclusions

In this review, we elucidated the successes, challenges, and shortcomings of CAR-T cell therapies with a major focus on key innovations and the pros and cons of CAR designs and strategies to overcome the obstacles in this emerging field (Table 1). CAR-T therapy has undoubtedly obtained significant success in the treatment of haematological tumours, and many therapies are now being actively translated into products. As previously reported, the difficulties of CAR-T therapy mainly lie in managing solid tumours, especially in the balance between the reduction in on-target/off-tumour toxicity and T cell persistence in the immunosuppressive TME [24]. Several strategies that can enhance the effectiveness of CAR T therapy for solid tumours are worth exploring. For target selection, higher tumour coverage could be achieved by using multi-target CAR T therapy, ensuring that more tumour cells can be eliminated, and mutant clonal escape is contained. However, higher coverage may also lead to more off-target cytotoxicity. Therefore, a careful and balanced evaluation is needed before clinical intervention. This “double trouble” makes it clear that applying CAR-T cells to treat solid tumours and inducing endogenous tumour-specific immune responses through CAR-T cell therapy should be a more powerful strategy. This concept has been confirmed in many preclinical studies and clinical cases [139,231,232]. Immunotherapy represents a benchmark in cancer research, and T cell-centred immunity mechanisms have gradually been revealed and exploited to create efficacious therapies like CAR-Ts. Along with advances in genetic engineering, it has become possible to exploit the immune system creatively and consciously to fight against cancer and other diseases. Modified CAR-T cells may represent a more effective alternative when natural T cells cannot effectively control cancer. What is undoubtedly needed is an improved understanding of the biological characteristics of T cells and of the different pathological contexts to design the best-engineered T cells to act and function in the specific setting and, hopefully, for a wide range of conditions. The frontier now is to bring into clinical assessment the various approaches and CAR designs (Figure 2) developed by scientists worldwide, for which we provided a comprehensive, updated, and thorough description here, looking for ever more adaptable, safer, and effective CAR-T therapies for cancer patients.

## Figures and Tables

**Figure 1 ijms-25-12201-f001:**
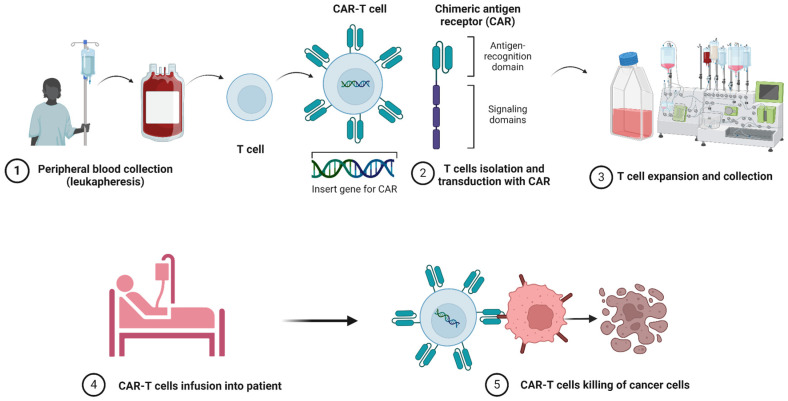
Schematic representation of the CAR-T cell production process. Autologous chimeric antigen receptor (CAR) T cell therapy, which is currently approved, works by modifying a patient’s own T cells ex vivo and then administering them back to the same person. The production process is carried out in accordance with Good Manufacturing Practice (GMP) guidelines and is overseen by multiple quality assessments. The first step in autologous CAR-T cell production is the isolation of T cells from the patient’s peripheral blood. Isolated T cells are then activated and genetically engineered via viral transduction to express the CAR molecule. CAR-T cells are subsequently expanded in culture to up to billions of cells. The modified CAR-T cells are typically cryopreserved and later re-infused into the patient to specifically target and eliminate cancer cells. Created in BioRender. Gaimari, A. (2022).

**Figure 2 ijms-25-12201-f002:**
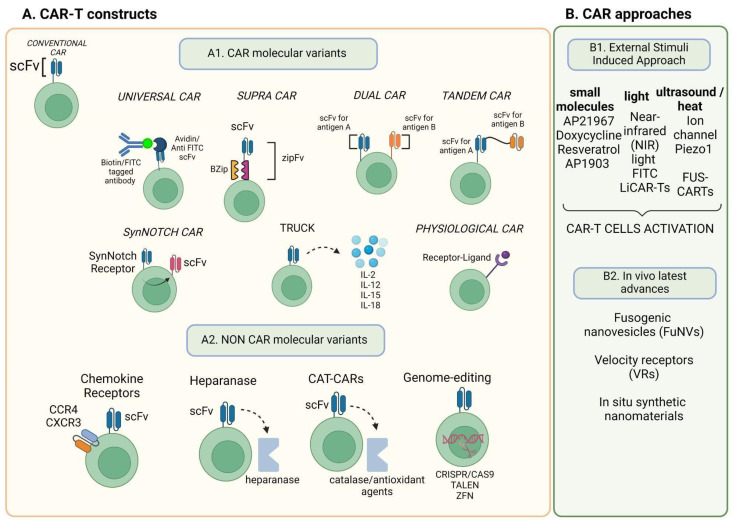
Schematic overview of CAR-T cell engineering strategies and recent innovations. The figure is divided into two main parts. Panel (**A**) shows various chimeric antigen receptor (CAR) T cell constructs, including UNIVERSAL CAR, DUAL CAR, SUPRA CAR, and Tandem CAR designs, highlighting structural modifications aimed at enhancing therapeutic specificity and efficacy. Additionally, non-design-based modifications, such as the incorporation of enzymes like heparanase or catalase, are shown as strategies to improve CAR-T cell performance. Panel (**B**) explores approaches driven by external stimuli, such as light, ultrasound, or small molecules (e.g., resveratrol, AP1903). The figure also highlights cutting-edge developments, including the use of synthetic nanomaterials and nanovesicles, which represent emerging tools to further refine CAR-T cell therapy, enhancing its safety and effectiveness in clinical applications. (scFv, single-chain variable fragment; FITC, fluorescein isothiocyanate; zipFv, tumour-targeting scFv adaptor; CCR4, C-C chemokine receptor type 4; CXCR3, C-X-C motif chemokine receptor type 3). Created in BioRender. Mazza, M. (2024), https://BioRender.com/p17t758 (accessed on 5 November 2024).

**Table 1 ijms-25-12201-t001:** Advantages and disadvantages of the different CAR designs. Key benefits and limitations across multiple CAR design strategies are summarised, indicating their therapeutic potential and clinical applications.

	Pros	Cons
**DUAL** **CAR** [43,44,45]	specific anti-tumour activity high persistence of CAR-T cells powerful cytokine production reduced “on-target/off-tumour” toxicity	clinical outcomes are unclear long patient follow-up lack of appropriate target antigens for some therapies (acute myeloid leukemia)
**TANDEM** **CAR** [34,46,47]	easier manufacturing procedures efficient protection against antigen escape increased safety and efficacy controllable mechanism reduced manufacturing costs reduced on-target/off-tumour toxicity persistence of cytotoxic activity with no significant exhaustion	manufacturing costs are reduced, but not disappeared reduced side effects, but not eliminated requiring both tumour antigens to function simultaneously
**UNIVERSAL** **CAR** [48,49,50,51,52,53,54,55,56]	leukapheresis is not required higher uniformity healthy T cells cheaper short manufacturing periods no limitation in cell sources prevention of GVHD by knocking out TCR	long-term follow-up limited penetration in tumour sites exhaustion side effects (CRS, on-target/off-tumour toxicity)
**SUPRA** **CAR** [57,58,59]	multi-targeted and programmable CAR high specificity for cancer cells targeting multiple antigens reduced tumour escape reduced relapse CRS controlled and reduced	required presence of more than one antigen complex manufacturing CRS is reduced but not absent
**SYNNOTCH** **CAR** [34,42,60,61,62,63,64]	flexible structure and higher accuracy in antigen recognition with respect to traditional CARs versatile applicability lower possibility of side effects production of cytokines possibility to be effective in solid tumours exhaustion is reduced more specific tumour killing	side effects are still present exhaustion can occur
**TRUCKS** [9,65,66,67]	TME conditioning via cytokines possibility to be effective against solid tumours more persistence in the tumour stroma	side effects still present no complete efficacy on solid tumours
**PHYSIOLOGICAL** **CAR** [34,68,69,70,71,72]	a ligand is used as an extracellular antigen-binding domain more potent cytotoxicity and less immunogenicity compared to conventional CARs high production of IFNg and cytokines programmable, effective, and specific	immunogenicity is only reduced cytotoxicity is not always effective

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
