# Peer review of "Significant Advancements and Evolutions in Chimeric Antigen Receptor Design"

_ijms, 2024, doi:10.3390/ijms252212201_

Round 1

Reviewer 1 Report

Comments and Suggestions for Authors

The authors present review on advances and challenges of different CAR-T cells. The authors present summary of different designs of CARs such as tandem CARs, SUPRA CARs, Switch-On/OFF CARs, Universal CARs. The authors describe interesting approaches to regulate CARs with light, ultrasound and other approaches that increase safety of CAR-T cells. There are several comments to improve manuscript.

1. In introduction, 6 CARs are mentioned approved by FDA but only 5 are described.

2. Table 1, the font is too small and difficult to read

3. The Summary Figure is required to show all mentioned designs for the reader to have easier understanding of each approach.

Author Response

Comments 1. In introduction, 6 CARs are mentioned approved by FDA but only 5 are described.

Response 1: We thank the Reviewer for noticing this issue. We have modified the text (highlighted in yellow, lines 131-136) by mentioning also the Carvykti® therapy.

Comments 2. Table 1, the font is too small and difficult to read

Response 2: We thank the Reviewer for helping us increase the readability of our work. We have adjusted the table, increased the font size, and included specific references for the different categories of CARs.

Comments 3. The Summary Figure is required to show all mentioned designs for the reader to have easier understanding of each approach.

Response 3: We acknowledge the Reviewer for this constructive suggestion. We have in fact reframed the figure, improving the clarity, and expanding the legend with a more detailed description of the different CARs approaches.

Reviewer 2 Report

Comments and Suggestions for Authors

Dear Authors,

Thank you for the review. Firstly, please verify if the review aligns well with the journal’s scope. To improve the paper, it's essential to add references for key points, specifically for lines 730-731 and 745, to enhance the credibility and depth of each claim. Expanding content in sections such as 3.3 and 3.4 will offer a more comprehensive understanding, particularly by including detailed explanations and comparative analyses related to heparanase production in CAR T cells.

Additionally, mentioning the software used for figure creation, like GraphPad Prism, Adobe Illustrator, or BioRender, either in a general note or within figure captions, will improve transparency. Figures, especially Figure 2, require refinement to enhance clarity and visual communication. A suggestion for Figure 2 is to split it into two parts: part (a) to depict different CAR T-cell constructs and part (b) to illustrate various therapeutic approaches or mechanisms. Further, adding annotations to the table will enhance overall readability and coherence. For the current table, which is a three-line format, it’s essential to include references indicating the sources of the pros and cons associated with different CAR T constructs.

Comments on the Quality of English Language

GOOD

Author Response

Comments 1: Thank you for the review. Firstly, please verify if the review aligns well with the journal’s scope. To improve the paper, it's essential to add references for key points, specifically for lines 730-731 and 745, to enhance the credibility and depth of each claim.

Response 1: We have improved the text as suggested by the reviewer (highlighted in yellow, from line 801  to line 854), in paragraph 3 and 3.1

Comments 2: Expanding content in sections such as 3.3 and 3.4 will offer a more comprehensive understanding, particularly by including detailed explanations and comparative analyses related to heparanase production in CAR T cells.

Response 2: We have expanded these sections as suggested by the reviewer , by describing heparanase and metalloproteases activity (highlighted in yellow, from line 905 to line 928), and incorporating a thorough elucidation of the immunosuppressive microenvironment and T-cells infiltration during CAR therapies, with a focus on  PD-1/PD-L1 role (highlighted in yellow, from line 934 to line 986).

Comments 3: Additionally, mentioning the software used for figure creation, like GraphPad Prism, Adobe Illustrator, or BioRender, either in a general note or within figure captions, will improve transparency. 

Response 3: We thank the Reviewer for pointing this out. We have mentioned the software used for figure preparation, along with the unique figure URL, directly in the figures’ legend.

Comments 4: Figures, especially Figure 2, require refinement to enhance clarity and visual communication. A suggestion for Figure 2 is to split it into two parts: part (a) to depict different CAR T-cell constructs and part (b) to illustrate various therapeutic approaches or mechanisms. 

Response 4: We appreciate the Reviewer for this valuable recommendation. As suggested, we have reframed the figure, dividing it in two parts to improve clarity. We also expanded the legend with a more detailed description of the different CARs approaches.

Comments 5: Further, adding annotations to the table will enhance overall readability and coherence. For the current table, which is a three-line format, it’s essential to include references indicating the sources of the pros and cons associated with different CAR T constructs.

Response 5: We thank the Reviewer for helping us increase the readability of our work. We have adjusted the table, increased the font size, and included specific references for the different categories of CARs.

Round 2

Reviewer 2 Report

Comments and Suggestions for Authors

thank you for revision